# Flagellar synchronization through direct hydrodynamic interactions

**Douglas R Brumley[1,2†], Kirsty Y Wan[1†], Marco Polin[1,3], Raymond E Goldstein[1]***

[1]Department of Applied Mathematics and Theoretical Physics, University of Cambridge, Cambridge, United Kingdom; [2]Department of Civil and Environmental Engineering, Massachusetts Institute of Technology, Cambridge, United States; [3]Department of Physics, University of Warwick, Coventry, United Kingdom

**Abstract** Flows generated by ensembles of flagella are crucial to development, motility and sensing, but the mechanisms behind this striking coordination remain unclear. We present novel experiments in which two micropipette-held somatic cells of *Volvox carteri*, with distinct intrinsic beating frequencies, are studied by high-speed imaging as a function of their separation and orientation. Analysis of time series shows that the interflagellar coupling, constrained by lack of connections between cells to be hydrodynamical, exhibits a spatial dependence consistent with theory. At close spacings it produces robust synchrony for thousands of beats, while at increasing separations synchrony is degraded by stochastic processes. Manipulation of the relative flagellar orientation reveals in-phase and antiphase states, consistent with dynamical theories. Flagellar tracking with exquisite precision reveals waveform changes that result from hydrodynamic coupling. This study proves unequivocally that flagella coupled solely through a fluid can achieve robust synchrony despite differences in their intrinsic properties.

**\*For correspondence:**
R.E.Goldstein@damtp.cam.ac.uk

[†]These authors contributed equally to this work

**Competing interests:** The authors declare that no competing interests exist.

**Reviewing editor**: W James Nelson, Stanford University, United States

## Introduction

Despite the elegance and apparent simplicity of the eukaryotic flagellum and its shorter ciliary version, the collective motions exhibited by groups of these organelles and the resultant fluid flows are far from trivial. For example, the unicellular biflagellate alga *Chlamydomonas reinhardtii* executes diffusive 'run-and-turn' locomotion (*Goldstein et al., 2009*; *Polin et al., 2009*) through stochastic switching between synchronized and unsynchronized swimming gaits—a process which could enhance searching efficiency and assist in the avoidance of predators (*Stocker and Durham, 2009*). Ensembles of cilia and flagella exhibit stunning temporal coordination, generating flows that transport mucus and expel pathogens (*Button et al., 2012*), establish the left-right asymmetry in developing mammalian embryos (*Nonaka et al., 2002*), and transport ova in human fallopian tubes (*Lyons et al., 2006*).

The origin of flagellar synchronization has been the subject of intense theoretical investigation for many decades. One of the earliest experimental results was Rothschild's qualitative observation (*Rothschild, 1949*) that the flagella of bull spermatozoa tend to synchronize when they swim close to one another, coupled only through the fluid surrounding them. Much more recent observations of self-organised vortex arrays of swimming sea urchin spermatazoa near surfaces (*Riedel et al., 2005*) provide further evidence for synchrony mediated purely by hydrodynamic coupling. Motivated by Rothschild's observation, Taylor (*Taylor, 1951*) developed a mathematical model in which two laterally infinite, inextensible sheets with prescribed sinusoidal travelling waves of transverse deformation interact with each other through a viscous fluid. He found that the rate of viscous dissipation is minimised when the two sheets are in phase. While minimisation of dissipation often holds in real physical systems, it is not in general a fundamental principle from which to deduce dynamical processes. Rather, an explanation for synchronization should capture the forces and torques associated with

**eLife digest** Sperm cells, as well as many bacteria and algae, propel themselves using whip-like appendages called flagella. Similar, shorter structures called cilia are also found on the surface of many cells, where they perform roles such as moving liquids over the cell.

Each cilium or flagellum beats at its own characteristic rhythm, but there are many situations where cilia or flagella must synchronize their beating with other nearby cells. For example, an egg cell is swept along the Fallopian tube by the coordinated beating of the cilia lining the tube. Bull sperm cells are also known to synchronize the beating of their flagella when swimming close to each other.

It has been suggested that the movement of the fluid surrounding the beating flagella could be the source of this synchronization. Experiments have produced results that match up with mathematical models describing this fluid movement. However, these experiments have often been designed in ways that didn't fully exclude other possible sources of synchronization, such as chemical signalling, or—for flagella located on the same cell—a physical connection between the flagella.

To overcome this shortcoming, Brumley et al. used high-speed imaging to watch the flagella of cells of *Volvox carteri*—a species of green alga—that were separated so that they could only communicate through the movement of the fluid around them. The flagella were still able to synchronize their beating, even when the two flagella naturally beat at substantially different rates.

The distance between the flagella affects how well the beating synchronizes. When close together, the flagella can lock into the same rhythm for thousands of beats. However, as they move further apart, random biochemical fluctuations within the cells reduce the extent to which the flagella can synchronize.

The flagella can also synchronize so that they move in the same direction at the same time, or in opposite directions, depending on how they are oriented relative to each other. Moreover, the results confirm that the fluid flow produced by a beating flagellum is sufficient to synchronize the beating of other nearby flagella.

the underlying molecular motors that drive flagella, their elasticity, as well as the viscosity of the surrounding fluid.

Since Taylor's work a myriad of increasingly complex models of flagellar synchronization have been proposed. Hydrodynamically coupled filaments or chains with various internal driving forces exhibit a general tendency towards synchrony (*Machin, 1963*; *Gueron et al., 1997*; *Guirao and Joanny, 2007*; *Yang et al., 2008*; *Elgeti and Gompper, 2013*). At the same time, minimal models of coupled oscillators in viscous fluids (*Vilfan and Jülicher, 2006*; *Niedermayer et al., 2008*; *Uchida and Golestanian, 2011*, *2012*; *Brumley et al., 2012*) offer great insight into the emergence of metachronal coordination. Such models have been investigated experimentally with light driven microrotors (*Di Leonardo et al., 2012*), rotating paddles (*Qian et al., 2009*) and colloids in optical tweezers (*Kotar et al., 2010*), and have also given rise to interpretations of the synchrony and coupling interactions between pairs of flagella of the model alga *Chlamydomonas* (*Goldstein et al., 2009*).

Although experimentally-derived coupling strengths between micropipette-held *Chlamydomonas* flagella are consistent with predictions based on direct hydrodynamic coupling (*Goldstein et al., 2011*), it has been proposed (*Friedrich and Jülicher, 2012*; *Geyer et al., 2013*) instead that this coupling is too weak to overcome noise, and that residual motion of elastically-clamped cells could play a role in synchronization. The recent observation (*Leptos et al., 2013*) of antiphase synchronization in a non-phototactic mutant of *Chlamydomonas* points as well to the possible role of *internal* mechanical coupling between flagella. Clearly, examining the synchronization between flagella on a single cell it is difficult to establish with certainty the origins of the coupling mechanism due to the likely presence of biochemical and elastic couplings of as yet unquantified strength between flagella.

In order to disentangle the hydrodynamic from the intracellular contributions to flagellar synchronization we conducted a series of experiments in which two physically separated flagellated cells, which exhibit distinct intrinsic beating frequencies in isolation, are coupled solely and directly through the surrounding fluid. These experiments can be viewed as natural generalisations of earlier work in which vibrating microneedles (*Okuno and Hiramoto, 1976*) or micropipettes (*Eshel and Gibbons, 1989*) are used to modulate and entrain the beating of a single sperm flagellum. Owing to the natural

distribution of beating frequencies of the flagella of its surface somatic cells, the colonial alga *Volvox carteri* is ideally suited to this purpose. Each somatic cell possesses two flagella which beat in perfect synchrony, facilitating their treatment as a single entity, henceforth referred to as the *flagellum*. Somatic cells were isolated from adult *Volvox* colonies and held with micropipettes at a controllable separation *d* (*Figure 1A,B*). The spatial and orientational degrees of freedom associated with this configuration enabled comprehensive analysis over a wide range of hydrodynamic coupling strengths. We found that closely-separated pairs of cells can exhibit robust phase-locking for thousands of beats at a time, despite a discrepancy in their intrinsic frequencies of as much as 10%. Both in-phase and antiphase configurations were observed, depending on the alignment of the directions of flagellar propulsion. Furthermore, with increasing interflagellar spacing we observed for each flagellum a marked change in the beating waveform, a key finding that lends support to models of synchronization that rely on waveform compliance to achieve phase-locking.

## Results

### One cell

We begin by characterising the flow generated by a single beating flagellum. Despite the fact that a flagellum is a spatially-extended object with considerable internal dynamics, it has become clear in recent years that the flow fields generated by its beating may be described, on suitable length scales, in terms of geometrically simpler force distributions. In the simplest case, often used in models of

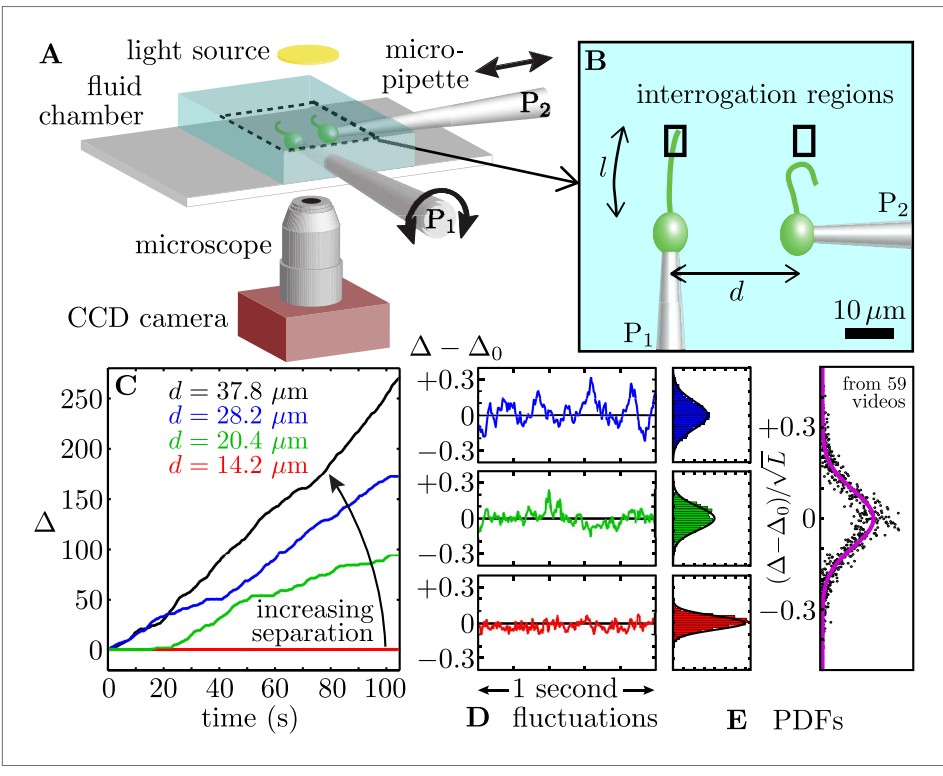

**Figure 1.** Synchronized pairs of beating flagella. (**A**) Experimental apparatus and (**B**) cell configuration. (**C**) Extracted phase difference $\Delta = (\varphi_1 - \varphi_2)/2\pi$ at four different interflagellar spacings, as indicated. These separations correspond to scaled spacing $L = d/l$ of 0.85, 1.22, 1.69, and 2.27. (**D**) fluctuations during phase-locked periods around the average phase lag, $\Delta_0$, and (**E**) the fluctuations' probability distribution functions (PDFs), each cast in terms of the rescaled separation-specific variable $(\Delta - \Delta_0)/\sqrt{L}$. Solid lines represent Gaussian fits. Further details of the phase extraction procedure can be found in *Figure 1—figure supplement 1*. Samples of the four processed videos corresponding to the cells in *Figure 1C* are shown in *Video 1*.

The following figure supplements are available for figure 1:

**Figure supplement 1**. Phase extraction.

synchronization (*Vilfan and Jülicher, 2006*; *Niedermayer et al., 2008*; *Uchida and Golestanian, 2011*), that would be just a single sphere tracing out a closed orbit in space under the action of internal driving forces. The time-averaged flow around the flagellum would then be approximated by the flow from a point force at a suitable average location. A point force $F$ exerted on a viscous fluid at a location $x_0$ produces a velocity field, known as a Stokeslet, of the form (*Blake and Chwang, 1973*) $u_i = F_j / 8\pi\mu r \left( \delta_{ij} + r_i r_j / r^2 \right)$, where the vector $r = x − x_0$, $r = |r|$ and $\delta_{ij}$ is the Kronecker delta. A recent study (*Drescher et al., 2010*) of freely swimming *Chlamydomonas* cells has shown that the time-averaged flow field is consistent in its magnitude and topology with a three-Stokeslet model (one for each flagellum and one for the cell body).

In our experiments, all flow fields were obtained using particle image velocimetry (PIV) (*Raffel et al., 2007*). *Figure 2A* shows the instantaneous flow field at four different times near a single cell, and it is clear that the magnitude (colour) and direction (vector field) of the flow vary during the cycle, as expected from the distinct power and recovery strokes. Examining the time-averaged velocity field (*Figure 2B*, obtained by averaging data from four cells), we see that for distances larger than 20 µm from the flagellar tip, both *upstream* (red) and *downstream* (blue) components of the flow obey a Stokeslet decay ($u \sim 1/r$) (*Figure 2C*). This trend is maintained over a range consistent with the distances sampled for our two-cell experiments (below).

Let us now examine more closely the time-dependent flows. *Figure 3A* and *Figure 3—figure supplement 1* show a fit of the instantaneous flow fields of each frame to a Stokeslet form, using the position $x_0$ and the magnitude and direction $F$ as fitting parameters. The results of this procedure are illustrated in *Figure 3A* as the average trajectory $\bar{x}_0(t)$ (the closed white curve) executed by the Stokeslet over approximately $10^3$ beats. *Figure 3B* shows $\bar{x}_0(t)$ (solid red line) together with a scatter plot of $x_0(t)$ from individual frames (red dots). The black arrows along the average cycle illustrate

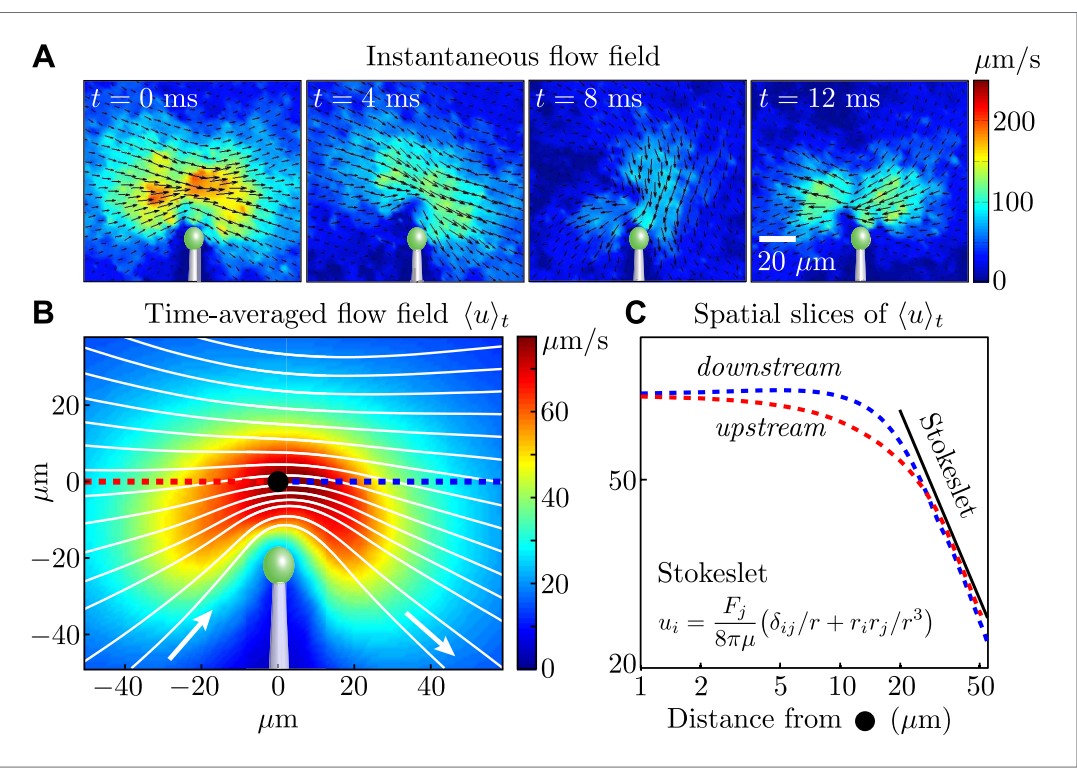

**Figure 2**. Measured flagellar flow field. (**A**) Time-dependent flow field for an individual cell measured using particle image velocimetry. Results are shown for the first half of the beating cycle. (**B**) Time-averaged flow field $\langle u \rangle_t = (1/\tau) \int_0^\tau |u(x,t)| dt$ (averaged across 4 cells with $\tau \sim 1000$ beats for each). The velocity magnitude (colour) and streamlines (white) are shown. (**C**) Velocity magnitude *upstream* (red) and *downstream* (blue) of the origin (black dot in **B**).

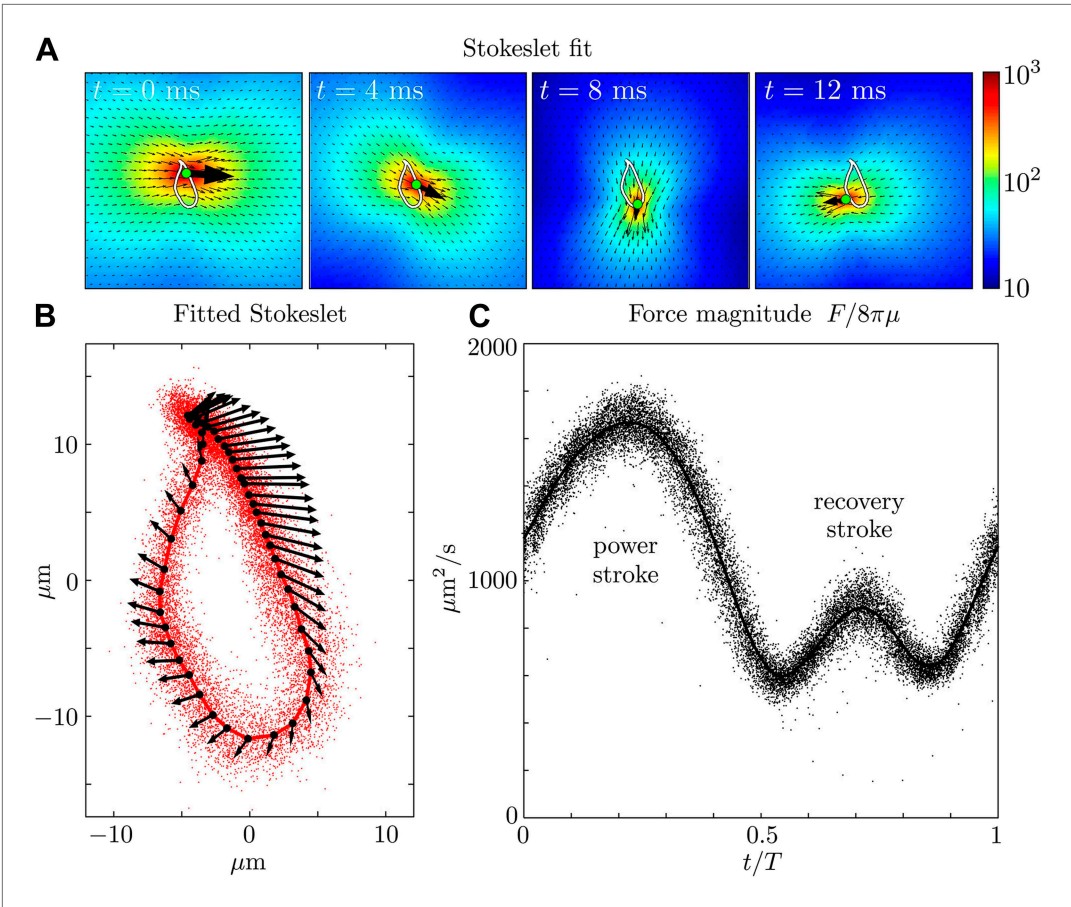

**Figure 3**. Force amplitude of flagellum. (**A**) Fitted instantaneous velocity field at various stages during the first half of one representative flagellar beat. (**B**) The fitted Stokeslet is shown at evenly-spaced times throughout the average flagellar beat cycle. The red dots indicate the Stokeslet position extracted from every frame. (**C**) Amplitude of the fitted point force as a function of time throughout the flagellar beat period $T$.

The following figure supplements are available for figure 3:

**Figure supplement 1**. Time-dependent flow fields.

the position, orientation and relative magnitude of the Stokeslet at evenly-spaced times along the *average cycle*. Importantly, the orientation of the point force does not coincide with its direction of motion, a feature to be expected given the anisotropic drag on the flagellum. *Figure 3C* shows the magnitude of the fitted Stokeslet over all beats. The amplitude of this force exhibits very strong periodic variations, and is approximated by $F(t)/8\pi\mu = A_0\left(1 + A_1\sin(2\pi t/T)\right)$ with $A_0 \simeq 1076\,\mu m^2/s$ and $A_1 \simeq 0.56$.

This determination of the magnitude of the effective Stokeslet describing the flow field around a cell can be compared with an estimate based on the observed motion of the flagellum itself. *Figure 4A* shows snapshots of a typical flagellum captured over a full beat cycle, superimposed at 2 ms intervals, together with measured instantaneous velocities along the filament. With resistive force theory (RFT), the results of the tracking procedure are used to derive estimates for the forces produced by the flagellum. First proposed by *Gray and Hancock (1955)*, RFT considers the anisotropic drag experienced by a long rod-like flagellum moving through a viscous fluid, and assumes that each unit segment of the flagellum experiences a local drag that is proportional to its local instantaneous velocity. The force density $f$ along the flagellum is approximated by

$$f = C_\perp u_\perp + C_\parallel u_\parallel, \tag{1}$$

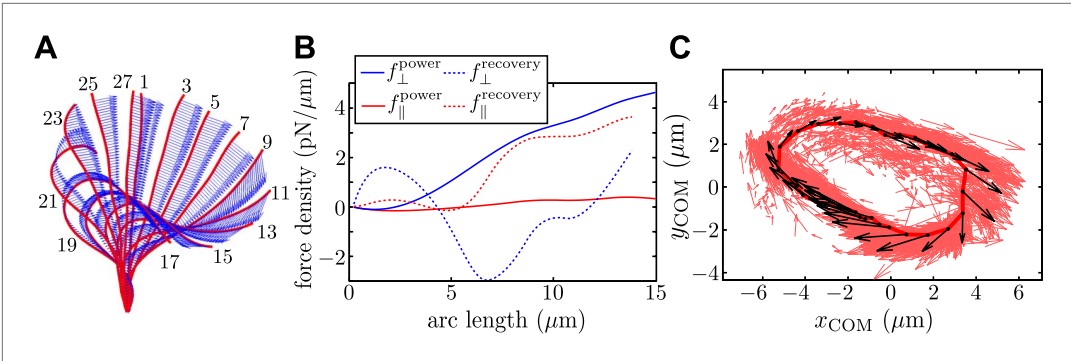

**Figure 4**. Resistive force theory analysis. (**A**) Instantaneous velocity distribution along the flagellum during one complete beat cycle (indexed by frame number, imaged at 1000 fps). (**B**) Components of integrated force density produced by a flagellum executing characteristic power and recovery strokes, as a function of arclength along the flagellum measured from the basal to the distal end. (**C**) Integrated vector forces $F(t)$ shown localised at centre-of-mass coordinates $x(t)$ (red: per frame, black: averaged over $O(10^3)$ frames), evolve cyclically around an average trajectory. The average value is $|\mathbf{F}|/8\pi\mu \sim 1910 \ \mu m^2/s$.

which is readily computable from experimental data, where the constants of proportionality, $C_\perp$ and $C_\parallel$, are the normal and tangential resistance coefficients respectively. We chose $C_\perp$ and $C_\parallel$ according to the classic model of *Lighthill (1975)*, with $C_\perp = 4\pi\mu / \left( \ln(0.18\lambda / a) + 0.5 \right)$ and $C_\parallel = 2\pi\mu / \left( \ln(0.18\lambda / a) - 0.5 \right)$, with an aspect ratio $\lambda/a = 80$.

The total instantaneous force $F(t)$ produced by the flagellum is given by $\int_0^l f(s,t)\,ds$, where $l$ is the total length of the flagellum and $s$ its arclength parameterisation. In *Figure 4B* we plot the normal (blue) and tangential (red) components of $f$, for characteristic power and recovery stroke waveforms (solid and dotted lines respectively). To construct a limit cycle representation of the cyclic force variation, we define an effective centre-of-mass for the flagellum, $x(t) = \sum_i^N \left( |f_i| \ x_i(t) \right) / \sum_i^N \left( |f_i| \right)$, averaging over all $N$ discretised force vectors $f_i$ applied at points $x_i$ along the flagellum. *Figure 4C* depicts the trajectories of integrated force $F$ in this coordinate representation (red arrows). An average limit cycle representation (black arrows) is obtained from measurements taken from ~100 beats: resultant force directions are seen to vary continuously along the cycle. The RFT result overestimates the force production during the recovery stroke, where the assumption of locality breaks down. It is encouraging to see that the amplitude of this force is similar to the value calculated earlier using Stokeslet fitting, though it should be noted that these results correspond to two different cells.

## Two cells

To investigate the effect of hydrodynamic coupling on *pairs* of flagella, we captured pairs of cells and aligned them so that their flagellar beating planes coincided (*Figure 1A*). Videos of hydrodynamically interacting flagella were first processed by subtracting a 30 frame running average. Median filtering was undertaken using $3 \times 3$ pixels$^2$ regions. At each cell–cell separation $d$, we recorded flagellar dynamics over ~100 s, and extracted flagellar phases $\varphi_{1,2}$ from Poincaré sectioning of the dynamics (*Goldstein et al., 2009*; *Polin et al., 2009*) by monitoring the signal in respective interrogation regions (*Figure 1B*, *Figure 1—figure supplement 1*), so that the respective flagella passed through precisely once per beat. Recording the passage times between beats allowed reconstruction of the flagellar phase $\varphi_{1,2}$. The time-dependent interflagellar phase difference $\Delta(t) = (\varphi_1 - \varphi_2)/2\pi$ was used to characterise the synchronization properties of the two cells.

The measured phase difference $\Delta(t)$ is shown in *Figure 1C* for one pair of cells at four different spacings (see *Video 1*). We measured beat frequencies $\omega_1$ and $\omega_2$ for the two flagella in isolation, and define $\delta\omega = \omega_1 - \omega_2$ to be their intrinsic frequency difference. Calling $L = d/l$ the cell–cell separation normalised by the average flagellar length $l$ of each pair, *Figure 1C* shows that for $L \gtrsim 2$ hydrodynamic coupling is negligible and $\Delta(t)$ drifts approximately linearly with time depending on $\delta\omega/\omega$ (8.2% here). For intermediate values of $L$, the flagella exhibit short periods of synchrony interrupted by brief phase slips. However, when the same cells are brought closer to each other, they phase-lock for the entire

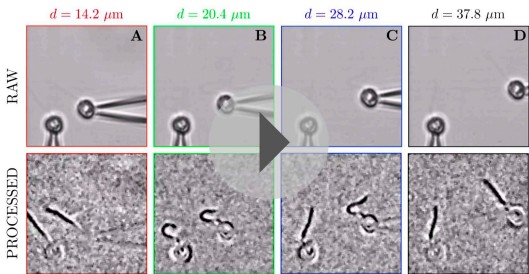

**Video 1**. A pair of hydrodynamically coupled flagella, observed at various cell–cell spacings. Original videos were recorded at 1000 fps, with processed representative segments (1000 frames each) replayed here at 25 fps.

duration of the experiment. This conclusively demonstrates that robust and extended flagellar synchronization can arise in physically separated cells purely through the action of hydrodynamics. For different pairs of cells ($n = 11$), a similar behaviour is observed.

Next we examine in detail the experimental time series $\Delta(t)$. Consider first the synchronous periods within the full time series of *Figure 1C*. Fluctuations about the phase-locked states $\Delta_0$ (*Figure 1D*) are Gaussian with a variance proportional to $L$, as seen by rescaling as $(\Delta - \Delta_0)/L^{1/2}$ (*Figure 1E*). Gaussian fluctuations suggest a description of the dynamics of $\Delta(t)$ based on a Langevin equation with an effective potential $V(\Delta)$ having a quadratic minimum at $\Delta_0$. We then write

$$\dot{\Delta} = -V'(\Delta) + \xi(t),$$

(2)

where $V(\Delta) = -\delta\nu\Delta + U(\Delta)$. The quantity $\delta\nu$ is the intrinsic frequency difference for the two phase oscillators, $U$ an effective potential which has period one in $\Delta$, and $\xi(t)$ is a Gaussian white noise term satisfying $\langle\xi(t)\rangle = 0$ and $\langle\xi(t)\xi(t')\rangle = 2T_{eff}\delta(t-t')$, where $T_{eff}$ is an effective 'temperature'. To leading order $U = -\varepsilon\cos(2\pi\Delta)$, where $\varepsilon$ is the interflagellar coupling strength. The observed dependence on $L$ of the distribution of $\Delta$ fluctuations is a natural consequence of *Equation 2* if $\varepsilon \propto 1/L$. We test this scaling below. Intraflagellar biochemical noise leads to stochastic transitions between adjacent minima of the tilted washboard potential $V(\Delta)$ (*Goldstein et al., 2009*; *Polin et al., 2009*). For each video, the autocorrelation of $\Delta$ is used to extract the model parameters $(\varepsilon, \delta\nu, T_{eff})$ as described previously (*Goldstein et al., 2009*; *Polin et al., 2009*).

Cells aligned so that their power strokes point in the same direction (as in many ciliates) exhibit *in-phase* (IP) synchrony ($\Delta_0 \simeq 0$), indicating a coupling strength $\varepsilon > 0$. Rotation of pipette $P_1$ (*Figure 1B*) by 180° so that the power strokes are opposed (as in the *Chlamydomonas* breaststroke) changes the sign of the coupling strength and gives rise to *antiphase* (AP) synchronization ($\Delta_0 \simeq 1/2$), in agreement with theory (*Leptos et al., 2013*). *Figure 5A* depicts the nondimensionalised coupling strength $\kappa = \varepsilon/\bar{\omega}$ for all experiments, where $\bar{\omega}$ is the average beat frequency across all experiments for a given pair of cells. The dependence on the interflagellar spacing $|\kappa| \propto L^{-1}$ is consistent with the intrinsic flagellar flow field presented in *Figure 2*. For both the in-phase and antiphase configurations, we fit $|\kappa| = k \times L^{-1}$ finding $k_{IP} = 0.016$ and $k_{AP} = 0.014$ respectively. At a given $L$, IP pairs exhibit on average a marginally stronger coupling than AP ones, possibly due to the fact that flagella in IP are on average closer together than in AP. The average values of the other model parameters are $\langle T_{eff}/\bar{\omega}\rangle = 0.005 \pm 0.003$ and $\langle\delta\nu/\bar{\omega}\rangle = 0.058 \pm 0.033$, with $\langle\bar{\omega}\rangle = 33.0$ Hz. As a cross-check, we can estimate directly the effective internal noise from the distribution of beating periods of separated cells, and find $\langle T_{eff}/\bar{\omega}\rangle = 0.002$, consistent with the value above.

The average *measured* flagellar frequency $\omega$ for the two cells in each experiment is shown in *Figure 5B*, nondimensionalised by the average value for each cell $\bar{\omega}_{cell}$ across different spacings. *Figure 5C* illustrates the measured frequency difference as a function of $L$. The data exhibit an apparent bifurcation near $L = 1$, beyond which phase drifting occurs over time. Integration of *Equation 2* in the absence of noise yields a predicted value for the observed frequency difference in terms of the model parameters: $\delta\omega/\delta\omega_{far} = \sqrt{1 - (2\pi\varepsilon/\delta\nu)^2}$, for $\varepsilon(L) < \delta\nu/2\pi$ and $\delta\omega/\delta\omega_{far} = 0$ otherwise. The orange curve in *Figure 5C* illustrates this prediction, calculated using the average extracted model parameters. In the presence of noise this sharp bifurcation becomes rounded and shifted (*Risken, 1989*), as shown in green in *Figure 5C*. It is evident that noise plays an important role in determining the observed location of the bifurcation point.

## Waveform characteristics

Although coupling is established purely through hydrodynamic interactions, the process of synchronization hinges on the ability of the flagella to respond differentially to varying external flows. For

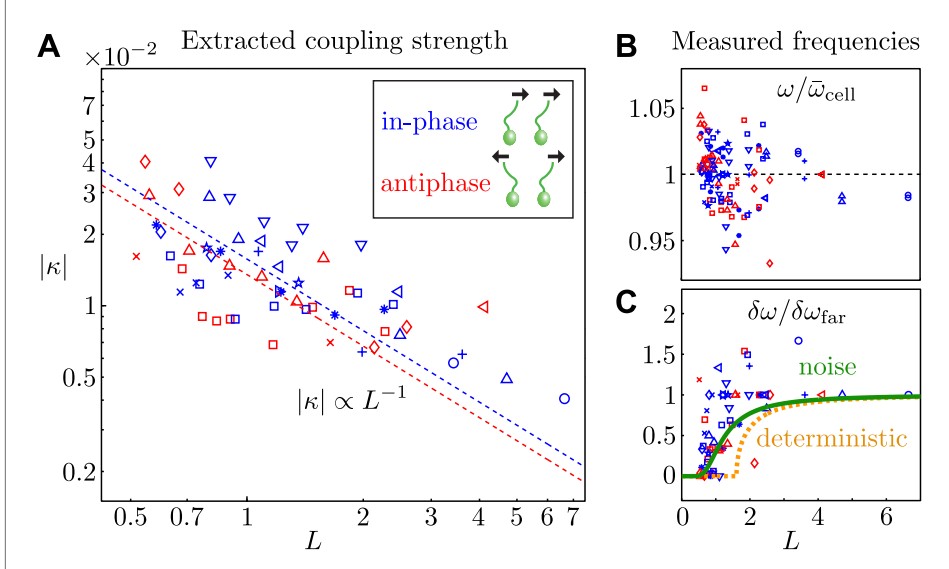

**Figure 5.** Coupling strength. (**A**) Dimensionless interflagellar coupling strength $\kappa = \varepsilon / \bar{\omega}$ as a function of the scaled spacing $L = d/l$ (log–log scale). The dotted lines represent fits of the form $|\kappa| = k \times L^{-1}$ with $k = 0.016$ (in-phase) and $k = 0.014$ (antiphase). (**B**) Measured beat frequency $\omega / \bar{\omega}_{cell}$ of each flagellum, nondimensionalised by the average value for that cell across several videos. (**C**) Measured frequency difference $\delta\omega / \delta\omega_{far}$ as a function of spacing $L$. The curves represent the predictions based on the average extracted model parameters in the absence (orange) and presence of noise (green). Symbols represent different pairs of cells, with the in-phase (blue) and antiphase (red) configurations shown.

sufficiently strong coupling, different cells can adopt a common phase-locked frequency through perturbing one another from their intrinsic limit cycles. Indeed, models of coupled flagella involving hydrodynamically coupled semiflexible filaments (*Gueron et al., 1997*; *Guirao and Joanny, 2007*; *Elgeti and Gompper, 2013*; *Yang et al., 2008*) show a tendency towards metachronal coordination, though the precise role that flexibility plays in facilitating synchrony is unknown. Minimal models in which spheres are driven along flexible trajectories (*Brumley et al., 2012*; *Niedermayer et al., 2008*) reveal that deformation-induced changes in the phase speed can facilitate synchrony. However, functional variations in the intrinsic flagellar driving forces could lead to synchrony even for fixed beating patterns (*Uchida and Golestanian, 2011*, *2012*).

Through dynamic tracking (*Wan et al., 2014*), we followed the evolution of the flagellar waveforms for several thousand consecutive beats. One example is shown in *Figure 6A* where the extracted waveform is shown at various stages through the beating cycle, overlaid onto logarithmically-scaled residence time plots. The same pair of flagella is compared at close and far cell–cell separations (7.3 μm and 72.2 μm respectively). In order to characterise flagellar waveform changes as the cells are brought closer together, we define three angles $x_a$, $x_b$, $x_c$ (radians) with respect to the cell body axis (*Figure 6B*). *Figure 6C* shows the temporal evolution of these angles for the right flagellum, corresponding to the close (red), intermediate (green) and wide (blue) separations. In particular, the most significant difference is observed in the $x_c$ component (distal part of the flagellum). Similar results are found for the other cell, indicating that the interaction is mutual. *Figure 6—figure supplement 1* and *Figure 6—figure supplement 2* illustrate the robustness of these results for multiple cells and different configurations. Taken together, the results in *Figure 6* demonstrate that accompanying the robust hydrodynamic phase-locking is a change in the flagellar waveform. For the first time, we have shown by systematically varying the cell–cell spacing that each flagellum can directly alter the beating profile of its neighbour simply through hydrodynamic interactions.

## Discussion

Understanding the mechanisms giving rise to robust phase-locking of flagella can be broken down into two distinct components, namely (i) identification of physical or chemical coupling between the flagella and (ii) characterisation of the response of each flagellum subject to these external stimuli.

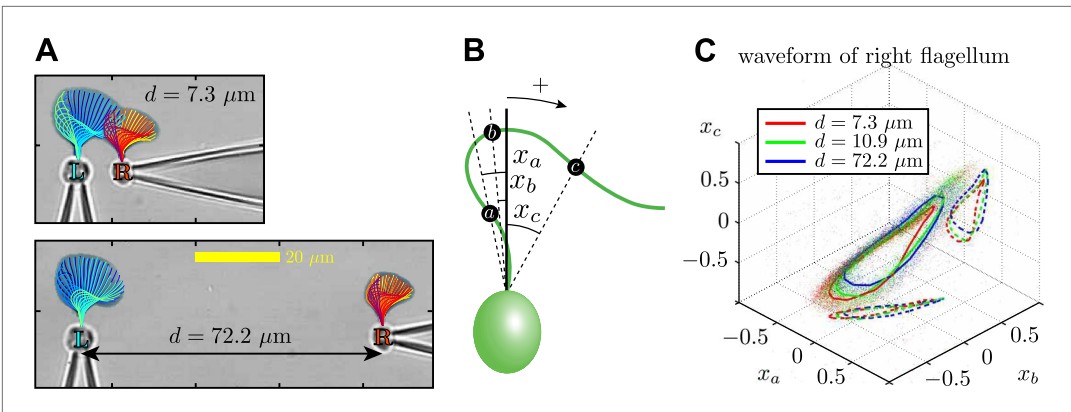

**Figure 6**. Waveform characteristics. (**A**) Logarithmically-scaled residence time plots of the entire flagella. The displayed waveforms correspond to 1 ms time intervals over several successive flagellar beats. (**B**) Angles $x_a$, $x_b$, $x_c$ (in radians) measured and (**C**) their characteristic 3D trajectories. Results are shown for the right flagellum, corresponding to three different interflagellar spacings. As the spacing $d$ is increased, the flagellar waveform exhibits a systematic change.

The following figure supplements are available for figure 6:

**Figure supplement 1**. Flagellar filaments are tracked for cells in the (A) antiphase state, as well as (B) the situation in which one of the cells does not possess a flagellum (dummy cell).

**Figure supplement 2**. Additional waveform data collected for 5 different cells in various geometric configurations.

Theoretical studies and experiments suggest that cell body rocking of freely swimming *Chlamydomonas* can induce synchrony (*Friedrich and Jülicher, 2012*; *Geyer et al., 2013*), and experimental investigations of such cells hint that hydrodynamic interactions between the flagella and cell body could be important for locomotion (*Kurtuldu et al., 2013*). At the same time, the synchronization properties of immobilised *Chlamydomonas* cells are generally consistent with models in which the flagella interact purely hydrodynamically (*Goldstein et al., 2009*, *2011*; *Polin et al., 2009*), although the observed antiphase synchronization of the *ptx1* mutant has been suggested (*Leptos et al., 2013*) to implicate intracellular coupling such as through elastic filaments at the basal bodies. In addition, intracellular calcium fluctuations in *Chlamydomonas* are known to affect the flagellar dynamics (*Yoshimura et al., 2003*; *Leptos et al., 2013*), so these studies cannot exclude the possibility that flagellar synchrony is regulated primarily by chemical or other non-mechanical means. Indeed, Machemer showed that membrane voltage affects the metachronal wave direction in *Paramecium* (*Machemer, 1972*).

In the present experiment, cells are held with micropipettes in order to preclude all chemical coupling of the two flagella other than the possible advection of molecules by the flow. The fact that the coupling strength for IP and AP pairs is almost identical despite the pronounced difference between the associated flows rules out coupling via chemical means. In these experiments hydrodynamics alone is responsible for the interflagellar coupling. Yet, this may be direct interaction between the flagella or indirect, through some residual motion of the cell bodies. If such motion does play a role in the observed synchronization, it is useful to estimate the effective spring constant associated with angular displacements of the cell. This can be done by estimating the hydrodynamical torques at one cell due to the flagellar beating of a nearby cell and using an experimental bound on the observed angular displacement. The fluid speed $u$ at a distance $r$ from the origin due to a second cell whose flagellum exerts a (point) force of magnitude $F$ varies as $u \sim F/8\pi\mu r$, and when acting on the flagellum of the first cell will produce a torque scaling as $C_\perp u l^2/2$, where $l$ is the length of the flagellum. If we set this equal to the torque $k\alpha$ of a rotational spring, where $k$ is the spring constant and $\alpha$ the angular displacement, we obtain an estimate for $k$ from an upper bound $\alpha^*$ for the rotation. Using $l = 20\ \mu m$, $r = 5\ \mu m$, and $F/8\pi\mu \sim 2 \times 10^3\ \mu m^2/s$, we find $k > (3 \times 10^2/\alpha^*)$ pN·μm. A similar argument can be constructed by considering the vorticity at the central cell due to the flow from another, and yields the same result. We have performed experiments of exactly this type, in which we measured the rocking motion of a deflagellated

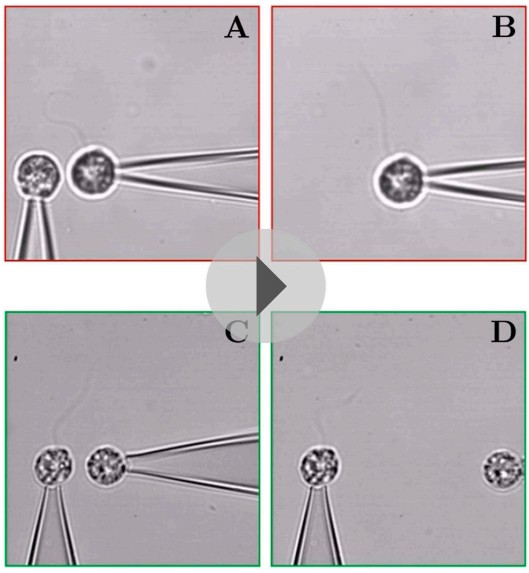

**Video 2**. Experiments in which one cell does not possess a flagellum. Figures (**A** and **B**) and (**C** and **D**) correspond to two different pairs of cells respectively. In each case, the flagellated cell exhibits some counterrotation during the beating, while the 'dummy cell' subject to the flow displays no visible rocking. Videos were recorded at 1000 fps and are replayed here at 25 fps.

cell placed in the flow field of a cell with a beating flagellum (**Video 2**), and found $\alpha^* \leq 0.01$ (about 0.5°). This strongly suggests $k > 3 \times 10^4$ pN·µm, which, by the calculations of *Geyer et al. (2013)*, is sufficiently large to suggest that rocking does not play a significant role in the synchronization observed in the present experiments.

Two spheres of radius $a$ in an unbounded fluid of viscosity $\mu$, driven along circular trajectories of variable radius (stiffness $\lambda$) are able to synchronize their motions through flow-induced changes in their respective phase speeds (*Niedermayer et al., 2008*). Calculating the spring stiffness $\lambda = R/l^3$ in terms of the flagellar bending rigidity $R$ and length $l$ yields the scaled coupling strength for this dynamical system, $\kappa_{\text{spheres}} = \left(27 \mu \pi a^2 l^2 \overline{\omega} / 2R\right) \times L^{-1}$. Estimating (*Niedermayer et al., 2008*) $a = 0.1$ µm and $R = 4 \times 10^{-22}$ Nm², and using measured values of the other parameters from the present experiment ($\langle \overline{\omega} \rangle = 33.0$ Hz and $\langle l \rangle = 19.9$ µm), we obtain $\kappa_{\text{spheres}} = 0.014 \times L^{-1}$. This minimal model, in which synchronization is facilitated through hydrodynamic interactions, compares favourably with the measured flagellar coupling strengths presented in **Figure 5**.

## Conclusions

The experimental study presented in this article reveals unambiguously the importance of hydrodynamics in achieving flagellar synchronization. Physical separation of the cells precludes any form of chemical or direct mechanical coupling, leaving hydrodynamic interactions as the only mechanism through which synchronization can occur. The process of phase-locking is extremely robust, with cells sufficiently close to one another exhibiting uninterrupted synchrony for thousands of consecutive beats. Accompanying this synchrony is a characteristic shift in the flagellar waveform. The extracted interflagellar coupling strength is consistent with hydrodynamic predictions and the measured flow fields generated by individual flagella. Additional experiments were undertaken using a uniflagellar mutant of the unicellular alga *Chlamydomonas*. Although its flagellum is shorter and its waveform is different to that of *Volvox*, we also observed hydrodynamic phase-locking in these experiments. Owing to the ubiquity and uniformity in the structure and function of flagella in various eukaryotic species, the results of the present study are expected to generalise to other systems, and may be of significant value for a wide range of theoretical models.

## Materials and methods

### Cell growth and imaging

*Volvox carteri* f. *nagariensis* (strain EVE) were grown axenically in Standard *Volvox* Medium (SVM) (*Kirk and Kirk, 1983*) with sterile air bubbling, in a growth chamber (Binder, Germany) set to a cycle of 16 hr light (100 µEm⁻²s⁻¹, Fluora, OSRAM) at 28°C and 8 hr dark at 26°C. Individual biflagellate cells were extracted from *Volvox* colonies using a cell homogeniser, isolated by centrifugation with Percoll (Fisher, UK), and inserted into a 25 × 25 × 5 mm glass observation chamber filled with fresh SVM. Cells were captured using micropipettes and oriented so that their flagellar beating planes coincided with the focal plane of a Nikon TE2000-U inverted microscope. Motorised micromanipulators (Patchstar, Scientifica, UK) and custom-made stages facilitated accurate rotation and translation of the cells. The flow field characterisation and pairwise synchronization analyses were imaged using a 40× Plan Fluor objective lens (NA 0.6). A higher magnification 63× Zeiss W Plan-Apochromat objective lens (NA 1.0) was used to conduct separate experiments for the waveform analysis. For each experiment, we recorded videos with a high-speed video camera (Fastcam SA3, Photron, USA) at 1000 fps under bright field illumination.

### One cell

Spatiotemporal analysis of the flow field associated with individual isolated cells was achieved through seeding the fluid with 0.5 μm polystyrene microspheres (Invitrogen, USA) at a volume fraction of $2 \times 10^{-4}$. We recorded ~30 s long videos, each one corresponding to approximately 1000 flagellar beats. The time-dependent velocity field was reconstructed using an open source particle image velocimetry (*Raffel et al., 2007*) toolbox for MATLAB (MatPIV).

### Two cells

For each pair of *Volvox* somatic cells, we investigated the synchronization properties as a function of interflagellar spacing. A number of videos were taken at various separations (varied non-monotonically). In many cases we also rotated the micropipettes (see *Figure 1B*) so that the flagella were beating in the same plane but opposite directions. There are two such 'antiphase' configurations possible, in which the flagella beat *towards* and *away* from one another respectively. Both of these states are referred to as *antiphase* in the extraction of parameters in *Figure 5* and *Figure 7*.

### Additional model parameters

The stochastic Adler equation was used to model the dynamics of $\Delta(t)$ as described in *Goldstein et al. (2011)*. *Figure 7A,B* show the amplitude $C_0$ of the autocorrelation function of $\Delta$ and the values of the average synchronous period $\tau_{sync}$. Fluctuations of the phase difference $\Delta$ about the synchronized states are well described by Gaussian distributions, with variances $C_0$ proportional to the interflagellar spacing $L$. The coupling strength $\varepsilon$ exhibits excellent agreement with the hydrodynamic predictions. *Figure 7C,D* show the dependence of the effective temperature $T_{eff} / \bar{\omega}$ and intrinsic frequency difference $\delta\nu / \bar{\omega}$ as a function of $L = d/l$ for every pair of flagella measured.

### Proximity to pipettes

In order to study the dynamics of hydrodynamically coupled flagella, the two cells were held using orthogonally-positioned glass pipettes. This geometry allowed us to investigate both in-phase and antiphase configurations for the same pair of cells, through the simple rotation of one pipette. At the same time, however, this meant that the two cells were held from different directions with respect to their flagella, and that one of the two pipettes was oriented along the direction of the flagellar power stroke, which is the main flow direction. This can cause two problems. Firstly, the flow field of a cell held by the side could be significantly different from that presented in *Figure 2*. Secondly, the holding pipettes could distort the scaling of the flagellar flow with cell–cell separation from the ~$1/r$ scaling

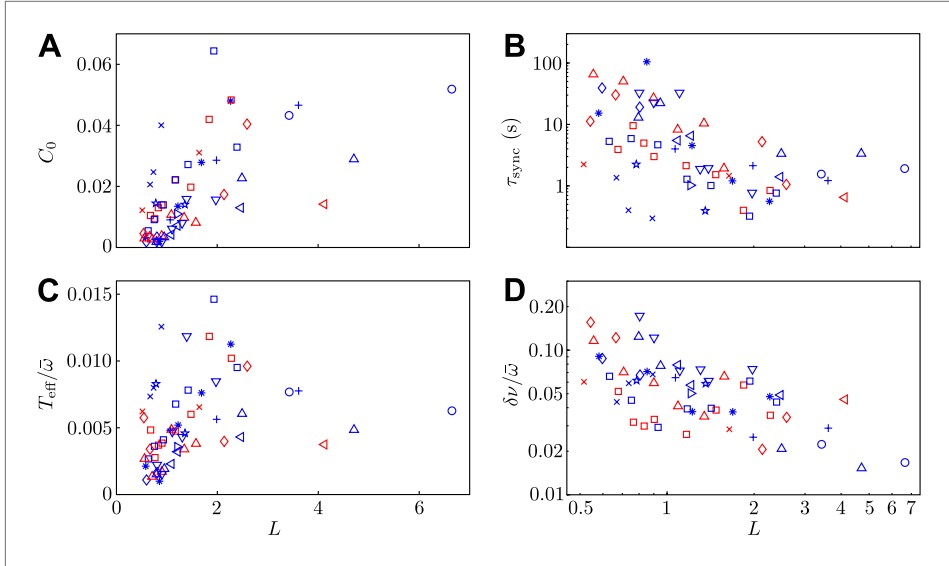

**Figure 7**. Model parameters. Two of the experimental observables (**A**) $C_0$ and (**B**) $\tau_{sync}$, and the two additional model parameters (**C**) $T_{eff} / \bar{\omega}$ and (**D**) $\delta\nu / \bar{\omega}$ are shown as functions of interflagellar spacing for all experiments conducted.

presented in **Figure 2C**. We investigated these problems with the series of experiments shown in **Figure 8**. One cell was held at its posterior pole by a pipette (**Figure 8A**) and the flow field measured. A second micropipette was then moved progressively closer, eventually to the point of contact with the cell (**Figure 8D**). It is clear that the second pipette affects the flow, but mostly in the region between the two pipettes.

Let us consider the region upstream of the cell (above the cell in **Figure 8**). For a cell held from the side, this is the region where the other cell will be. Here the flow is only minimally affected, with an average relative change between **Figure 8D,A** below 8%. A large contribution is represented simply by a ~7% decrease in flow speed. Taking this decrease into account, the average relative change is about 5%. As a result, these experiments allow us to consider the flow generated by a cell held from the side as identical to that generated by a cell held from the back, at least in the region of interest to our experiments. By comparing the flows for different positions of the second pipette, we can also quantify its effect on the flow field that would be experienced by the second cell. For each configuration of pipettes, this can be estimated as the relative difference between the unperturbed and the perturbed flows in the region where the flagella of the second cell would be, here considered to be a 10 × 10 μm² region 20 μm to the left of the tip of the incoming pipette. The difference ranges from ~5% to ~10% and ~13% for **Figure 8B–D** respectively (in the last case we choose a position approximately 10 μm below and 20 μm to the left of the pipette tip). These represent the typical error contributions from neglecting, as we have done in the text, the influence of the pipettes on the flows generated by the cells.

## Minimal model with variable forcing

We used the Stokeslet approximation to the flow field of an isolated cell in **Figure 3**, to test the effect of force modulation on synchronization within the class of minimal models which abstract the beating flagellum as a sphere driven along a closed orbit (**Niedermayer et al., 2008**; **Uchida and Golestanian, 2011**, **2012**). We simulated two spheres of radius $a = 0.75$ μm in an unbounded fluid of viscosity $\mu = 10^{-3}$ Pa·s, driven along coplanar circular orbits of radius $r_0 = 8$ μm by a force $F(\varphi)/8\pi\mu = A_0\left(1 + A_1\sin\left(\nu\varphi + \varphi_0\right)\right)$ tangential to the orbit, with $A_0 = 1076$ μm²/s and $A_1 = 0.56$. Notice that this corresponds to assuming that the point forces in **Figure 3B** are tangential to the cycle. The value $a = 0.75$ μm was chosen to ensure that the orbital frequency matched the mean value observed experimentally. The orbits were separated by $d = 20$ μm and had a radial stiffness with spring constant $\lambda$. The limit $\lambda \rightarrow \infty$ corresponds to rigid prescribed trajectories (holonomic constraint). For each value of $\lambda \in \left\{1\,\text{pN}/\mu\text{m}, 5\,\text{pN}/\mu\text{m}, \infty\right\}$ we ran five sets of simulations, corresponding to $\nu \in \left\{0, 1, 2\right\}$ and $\varphi_0 \in \left\{0, \pi/2\right\}$. Choosing $\nu = 2$ is equivalent to modulating the driving force with the experimental

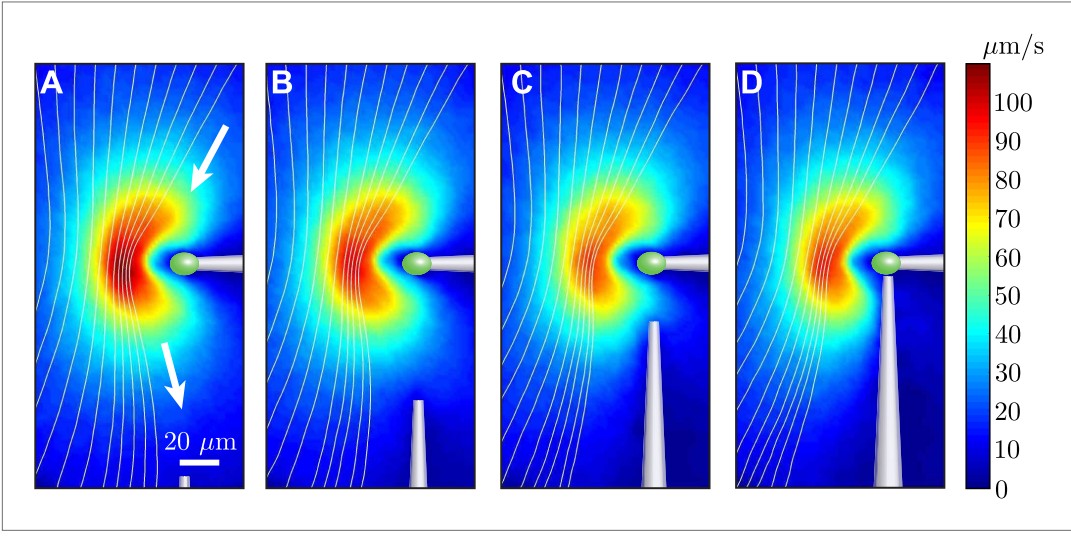

**Figure 8**. Effect of nearby pipette. The time-averaged flow field associated with one captured cell is measured as a second pipette slowly approaches. This demonstrates that the precise angle from which the cell is held by the micropipette has very little effect on the resultant flow field.

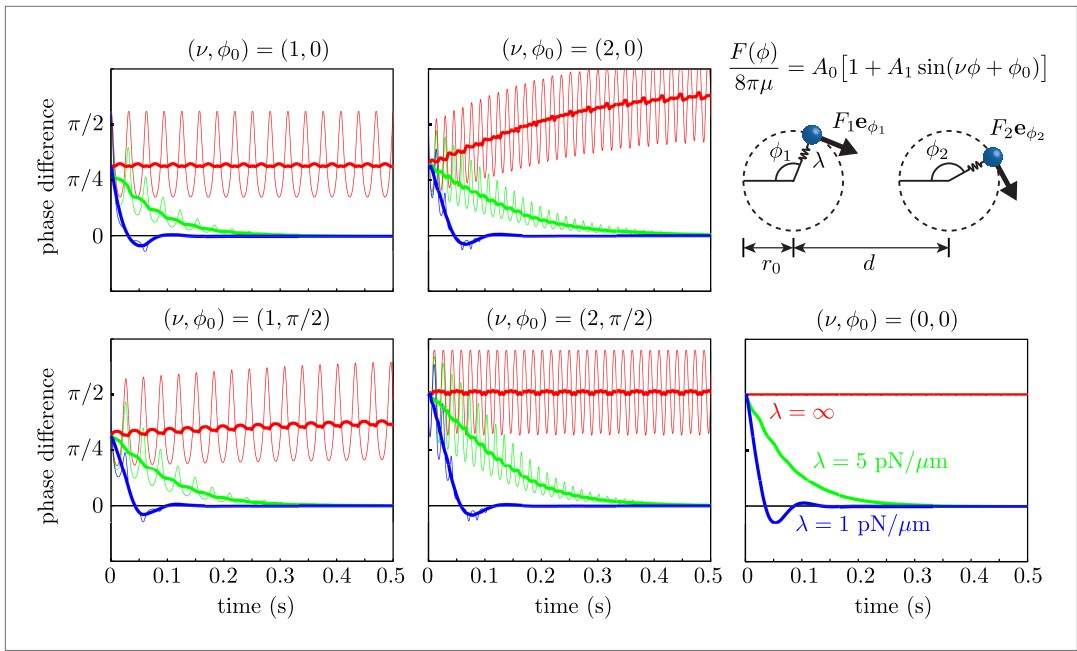

**Figure 9**. Effect of force modulation. Evolution of the phase difference $\delta = \varphi_1 - \varphi_2$ among two identical model oscillators, each composed of a sphere driven around a circular trajectory by a tangential driving force. The trajectories each possess a radial stiffness $\lambda$. Smaller values of $\lambda$ yield rapid convergence towards synchrony ($\delta = 0$), in a manner essentially independent of the functional form of the driving force. Parameters used are given by $a = 0.75$ μm, $r_0 = 8$ μm, $d = 20$ μm, $A_0 = 1076$ μm²/s and $A_1 = 0.56$.

amplitude but at a frequency double the experimental one. Although this is not what we observed, it is still interesting to consider, since in this configuration it is the frequency that contributes most to synchronization through force modulation.

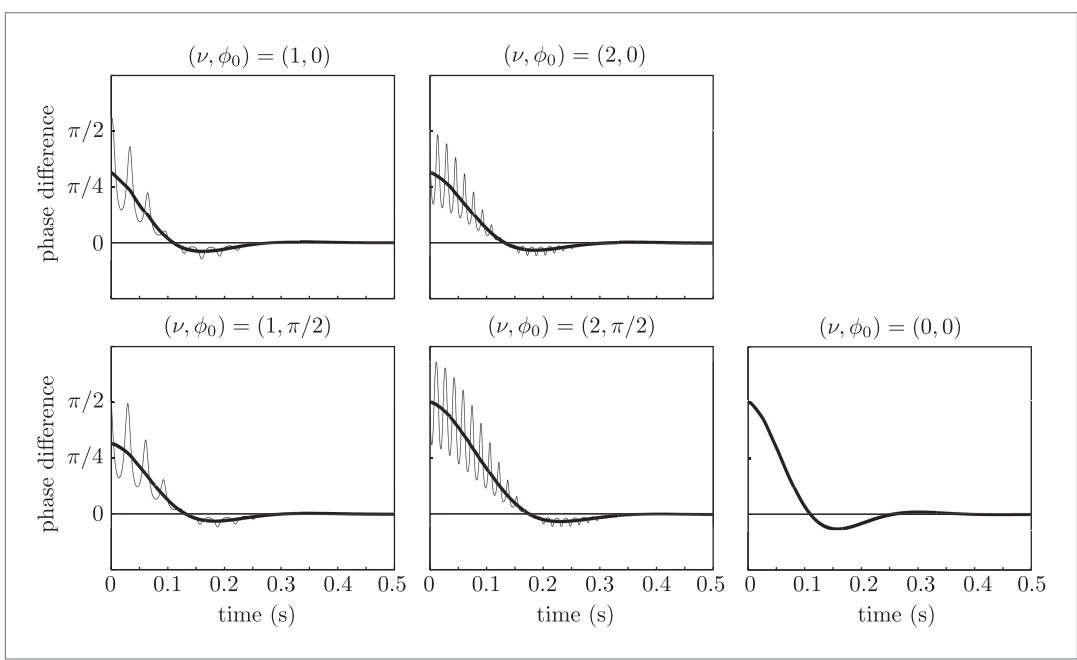

**Figure 10**. Effect of force modulation. Re-run of the simulations in *Figure 9* with properties inspired by real flagella.

As a consequence of the phase-dependent driving force, the geometric phase $\varphi_i$ of an *individual isolated* oscillator does not evolve at a constant rate in time. We thus chose to rescale the phase $\Phi = \Phi(\varphi)$ so that in the absence of hydrodynamic interactions, $\dot{\Phi} = 2\pi / T$ = constant. Both the *geometric* phase difference $\delta = \varphi_1 - \varphi_2$ (thin curves) and its *rescaled* value $\delta_{rescaled} = \Phi_1 - \Phi_2$ (thick curves) are shown for each simulation in *Figure 9*. These results show clearly that *within the boundary* of the model we are considering, the two oscillators synchronize through a coupling between hydrodynamic stresses and orbit compliance (*Niedermayer et al., 2008*) with no noticeable effect from force modulation.

Repeating the simulations with a stiffness derived from the flagellar bending rigidity as in the main text, $\lambda$ = 0.05 pN/μm, radius $a$ = 0.1 μm, and reducing the force amplitude to $A_0$ = 143 μm²/s to keep the revolution frequency at the experimental value, yields the results in *Figure 10*. Again, the synchronization is achieved only through interaction between hydrodynamic stresses and orbit compliance.

## Additional information

### Funding

| Funder | Grant reference number | Author |
| --- | --- | --- |
| European Research Council | Advanced Investigator Grant 247333 | Douglas R Brumley, Kirsty Y Wan, Marco Polin, Raymond E Goldstein |
| Wellcome Trust | Senior Investigator Award | Douglas R Brumley, Kirsty Y Wan, Raymond E Goldstein |
| Engineering and Physical Sciences Research Council | | Kirsty Y Wan, Marco Polin, Raymond E Goldstein |
| Human Frontier Science Program | | Douglas R Brumley |

The funders had no role in study design, data collection and interpretation, or the decision to submit the work for publication.

### Author contributions

DRB, KYW, MP, Conception and design, Acquisition of data, Analysis and interpretation of data, Drafting or revising the article; REG, Conception and design, Analysis and interpretation of data, Drafting or revising the article

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
