## [Decision Letter]

Thank you for sending your work entitled “Flagellar Synchronization Through Direct Hydrodynamic Interactions” for consideration at *eLife*. Your article has been favorably evaluated by Randy Schekman (Senior editor), a Reviewing editor, and 3 reviewers.

The Reviewing editor and the reviewers discussed their comments before we reached this decision, and the Reviewing editor has assembled the following comments to help you prepare a revised submission.

The manuscript by Brumley et al. reveals mechanisms of beat synchronization in pairs of flagella. This is biologically relevant as pairs of flagella serve as a minimal model system for synchronization in collections of flagella such as ciliary carpets in mammalian airways, oviduct epithelium, etc. whose coordinated beating in metachronal waves is important for efficient fluid transport. Secondly, this minimal system allows to link biology and physics in a quantitative way to generate mechanistic insight.

The work reports a key experiment; in fact this experiment should have been conducted long ago. By use of two spatially separated cells, any means of chemical coupling or direct mechanical coupling is effectively ruled out. This sets this particular experiment apart from previous studies.

The 3 reviewers and the Reviewing editor agreed that the manuscript is well-crafted and contributes to a timely topic. However, the authors should address the following significant points:

1) The analysis of waveform is elegant, but there is a lack of statistics or a way to compare results from multiple cell pairs in Figure 4, which just shows one example of a pair of cells. At the very least, several other examples need to be shown, perhaps as figure supplements, to show that the effect of distance on waveform isn't just some idiosyncrasy of this particular cell pair. What would be more ideal is if there is some way to parameterize the limit cycles so that their shape could be described by a single number or some small number of parameters. The cycles look teardrop shaped so maybe one could use a Pyriform (“pear shaped quartic”), or the Lemniscate of Bernoulli. Regardless of how it is accomplished, having some ability to compare results from different cells statistically is very important.

2) It is unclear whether there is a true bifurcation in the stochastic case. If delta_omega is defined as the expectation value of dphi1/dt-dphi2/dt, then delta_omega likely varies smoothly with epsilon, although it may indeed display a steep change beyond a certain value. A more stringent terminology is required.

3) Were different stages used for the two micropipettes to rule out any direct mechanical coupling between them? Otherwise, there is a concern that potential oscillations of the micropipettes caused by the flagellar oscillations could by relayed by such a coupling. One cilium is certainly too weak to generate such a strong vibration to excite the manipulator and then back to the other cilium. There is of course the potential that the stage itself is vibrating and then causes both of them to oscillate. Sperm cells have been held on vibrating micro needles, and vibrating micro needles were brought close to flagella (Eshel 1989 and Okuno 1976); from there the authors could potentially estimate how strongly the manipulator would have to vibrate.

4) Referencing previous published work: the authors should do a better job in giving credit to previous relevant work (which goes far back and another detailed literature search is advised). Giving corresponding credit to this previous work does not diminish the relevance of the present paper given its experimental execution and analysis.

a) A clear statement in the paper should be made that (pure) hydrodynamic synchronization between cilia/flagella has been demonstrated previously. The authors cite [33] in a manner that suggests that Taylor made pure theoretical arguments (which is actually true), but Taylor himself cites earlier experiments that describe the synchronization of sperm flagella.

b) It would be worth commenting on the dominating synchronization mechanism of the flagellar pair in single clamped Chlamydomonas One group (Goldstein et al. PRL 2009) proposed direct hydrodynamic interactions between the two flagella of Chlamydomonas as main synchronization mechanism, while others have argued that this synchronization mechanism might be too weak to overcome flagellar noise and that instead a residual motion of elastically clamped cells could accelerate flagellar synchronization: VF Geyer, F Jülicher, J Howard, BM Friedrich, PNAS 110, 18058–18063 (2013).

c) Riedel 2005 showed synchronization of sperm flagella, with a similar conclusion on the hydrodynamic origin of the coordination.

d) There have been also multiple papers where sperm cells have been held on vibrating micro needles, or vibrating micro needles were brought close to flagella, see for example Eshel 1989 and Okuno 1976 – and there are likely more.

e) The cited reference [9] combines theory and experiments, which should be mentioned. Also, the discussion of elastic contributions to flagellar synchronization should make reference to this cited reference, which first introduced alternative mechanisms of flagellar synchronization independent of direct hydrodynamic interactions.

---

## [Author Response]

*1) The analysis of waveform is elegant, but there is a lack of statistics or a way to compare results from multiple cell pairs in*
Figure 4*, which just shows one example of a pair of cells. At the very least, several other examples need to be shown, perhaps as figure supplements, to show that the effect of distance on waveform isn't just some idiosyncrasy of this particular cell pair. What would be more ideal is if there is some way to parameterize the limit cycles so that their shape could be described by a single number or some small number of parameters. The cycles look teardrop shaped so maybe one could use a Pyriform (“pear shaped quartic“), or the Lemniscate of Bernoulli. Regardless of how it is accomplished, having some ability to compare results from different cells statistically is very important*.

We have now expanded Figure 6 (previously Figure 4) to include several figure supplements that illustrate the robustness of the results on waveform deformations.

We agree with the referees that the issue of limit cycle parametrisation is interesting, and indeed we had already thought about it. Ideally, we would like to project out limit cycles on some basis set of curves, which could then be used to quantify the limit cycles’ deformations. We believe that it is the extent of these deformations that should be compared between cells, rather than the limit cycles themselves. Unfortunately, we did not find any convincing way to put this idea into practice. In particular it was not obvious to us what would constitute a good metric in this space of curves. We decided, therefore, to present only the raw deformations. It is our hope that with time these results will inspire the development of a technique to quantify the deformations.

*2) It is unclear whether there is a true bifurcation in the stochastic case. If delta_omega is defined as the expectation value of dphi1/dt-dphi2/dt, then delta_omega likely varies smoothly with epsilon, although it may indeed display a steep change beyond a certain value. A more stringent terminology is required*.

In the presence of noise the bifurcation is indeed rounded, although for the level of noise in our experiments the predicted rounding is small. We have now made this clear in the text.

*3) Were different stages used for the two micropipettes to rule out any direct mechanical coupling between them? Otherwise, there is a concern that potential oscillations of the micropipettes caused by the flagellar oscillations could by relayed by such a coupling. One cilium is certainly too weak to generate such a strong vibration to excite the manipulator and then back to the other cilium. There is of course the potential that the stage itself is vibrating and then causes both of them to oscillate. Sperm cells have been held on vibrating micro needles, and vibrating micro needles were brought close to flagella (Eshel 1989 and Okuno 1976); from there the authors could potentially estimate how strongly the manipulator would have to vibrate*.

The microscope used in these studies sits on a very high quality floating vibration isolation table in the quietest room of our microscopy suite, with the room air supply silenced and delivered through tiny perforations in the ceiling tiles, preventing any vibration-inducing jets of air. The two micropipettes were held on physically separated supports in either of two ways. In one, each micropipette is mounted on a separate multi-axis translation stage supported on a vibrationally-damped rod attached to the optical table but with having no physical contact with the microscope. In the second approach, one of those micropipettes is instead supported on a miniature translation stage mounted directly on the microscope body, while the other is still mounted on a separate manipulator. With all these pre-cautions we are very confident the observed flagellar dynamics is indeed intrinsic to the cells. We thank the referee for highlighting the earlier works on sperm cells and vibrating micro needles, which we have now referenced in the introduction.

*4) Referencing previous published work: The authors should do a better job in giving credit to previous relevant work (which goes far back and another detailed literature search is advised). Giving corresponding credit to this previous work does not diminish the relevance of the present paper given its experimental execution and analysis*.

*a) A clear statement in the paper should be made that (pure) hydrodynamic synchronization between cilia/flagella has been demonstrated previously. The authors cite*
[33]
*in a manner that suggests that Taylor made pure theoretical arguments (which is actually true), but Taylor himself cites earlier experiments that describe the synchronization of sperm flagella*.

Thank you for this comment. We are aware that Taylor’s work was indeed motivated by an observation of Rothschild that the flagella of closely-separated bull spermatozoa tend to be synchronized, and have now included this reference in the Introduction. We have further taken this opportunity to expand the Discussion in the Introduction about hydrodynamic mechanisms. It is important to note that neither Rothschild’s observations nor those of the more recent work we cite provided unequivocal proof of purely hydrodynamic synchronization. For example, a mechanism akin to cell body rocking may have played a role there as well.

*b) It would be worth commenting on the dominating synchronization mechanism of the flagellar pair in single clamped Chlamydomonas One group (Goldstein et al. PRL 2009) proposed direct hydrodynamic interactions between the two flagella of Chlamydomonas as main synchronization mechanism, while others have argued that this synchronization mechanism might be too weak to overcome flagellar noise and that instead a residual motion of elastically clamped cells could accelerate flagellar synchronization: VF Geyer, F Jülicher, J Howard, BM Friedrich, PNAS 110, 18058–18063 (2013)*.

We have modified the Discussion as requested.

*c) Riedel 2005 showed synchronization of sperm flagella, with a similar conclusion on the hydrodynamic origin of the coordination*.

We have now included this reference in the Introduction.

*d) There have been also multiple papers where sperm cells have been held on vibrating micro needles, or vibrating micro needles were brought close to flagella, see for example Eshel 1989 and Okuno 1976 – and there are likely more*.

These references have also been included in the revised Introduction.

*e) The cited reference*
[9]
*combines theory and experiments, which should be mentioned. Also, the discussion of elastic contributions to flagellar synchronization should make reference to this cited reference, which first introduced alternative mechanisms of flagellar synchronization independent of direct hydrodynamic interactions*.

Done.